# Airborne Electromagnetic and Radiometric Peat Thickness Mapping of a Bog in Northwest Germany (Ahlen-Falkenberger Moor)

**Bernhard Siemon [1],\* [ID], Malte Ibs-von Seht [1] and Stefan Frank [2],†**

[1] Federal Institute for Geosciences and Natural Resources (BGR), 30655 Hannover, Germany; Malte.Ibs-vonSeht@bgr.de

[2] State Authority for Mining, Energy and Geology (LBEG), 30655 Hannover, Germany; Stefan.Frank@thuenen.de

\* Correspondence: Bernhard.Siemon@bgr.de; Tel.: +49-511-643-3488

† Current address: Thünen Institute of Climate-Smart Agriculture, 38116 Braunschweig, Germany.

**Abstract:** Knowledge on peat volumes is essential to estimate carbon stocks accurately and to facilitate appropriate peatland management. This study used airborne electromagnetic and radiometric data to estimate the volume of a bog. Airborne methods provide an alternative to ground-based methods, which are labor intensive and unfeasible to capture large-scale (>10 km$^2$) spatial information. An airborne geophysical survey conducted in 2004 covered large parts of the Ahlen-Falkenberger Moor, an Atlantic peat bog (39 km$^2$) close to the German North Sea coast. The lateral extent of the bog was derived from low radiometric and elevated surface data. The vertical extent resulted from smooth resistivity models derived from 1D inversion of airborne electromagnetic data, in combination with a steepest gradient approach, which indicated the base of the less resistive peat. Relative peat thicknesses were also derived from decreasing radiation over peatlands. The scaling factor ($\mu_a = 0.28$ m$^{-1}$) required to transform the exposure rates (negative log-values) to thicknesses was calculated using the electromagnetic results as reference. The mean difference of combined airborne results and peat thicknesses of about 100 boreholes is very small (0.0 ± 1.1 m). Although locally some (5%) deviations (>2 m) from the borehole results do occur, the approach presented here enables fast peat volume mapping of large areas without an imperative necessity of borehole data.

**Keywords:** peatland; bog; fen; peat thickness; volume mapping; airborne electromagnetic survey; airborne gamma-ray survey; EM inversion

## 1. Introduction

Peatlands release greenhouse gases to the atmosphere, particularly if anthropogenic drainage and land use for agricultural, silvicultural, or horticultural purposes take place. In Germany, 2.8% (7.8 Mio. T CO$_2$-C-equivalents) of the total national greenhouse budget of 2006 came from peatlands and horticultural peat extraction and use [1]. Knowledge on peat volumes of peatlands (both peat bogs and fens) is essential to estimate carbon stocks accurately and to facilitate appropriate peatland management [2], as well as to support nature conservation policies.

Commonly used direct soil probing of peatlands is labor intensive and unfeasible to capture spatial information at landscape extents. Ground-based geophysical methods, such as ground-penetrating radar and geoelectrics, have been successfully applied at the surface to investigate peatlands due to their electrical parameter permittivity and conductivity (e.g., [3–6]), but they are also restricted to walk-in areas and not suitable for the investigation of large peatlands. Remote sensing and airborne geophysical methods, however, may help to overcome this challenge, but they have not

been investigated sufficiently so far. Remote sensing methods are able to provide information about the surface characteristics of a peatland, e.g., surface elevation, slope, topographic wetness index, or drainage system, but they require additional information to estimate the peat thickness [2]. From all the airborne geophysical methods successfully used for, e.g., mineral or groundwater exploration, airborne radiometric and electromagnetic surveys, in particular (e.g., [7]), may help to estimate peat thickness and extent. These and other methods for the digital mapping of peatlands are discussed in a comprehensive review recently published by Minasny et al. [8].

The estimation of peat thickness from radiometric data is difficult because the sources, i.e., the material beneath the peatland, and the absorbers, i.e., water-saturated peat, have to be taken into account. Generally, the source radiation is not known and has to be assumed. Furthermore, Beamish [9] showed that 0.5 m of water-saturated peat (80% porosity, density 0.1 g/cm$^3$) reduces about 90% of the source radiation and the attenuation heavily depends on the degree of water saturation. Attenuation also occurs in air, i.e., the higher the gamma-ray spectrometer the lower the count rates. Nevertheless, Keaney et al. [10] and others proposed using airborne gamma-ray (AGR) data for peat thickness mapping if the radiometric data were calibrated on known thickness values (from in situ measurements). Therefore, it is mandatory to combine the radiometric results with other methods. Gatis et al. [2], for example, combined radiometric and remote sensing (LiDAR) data to develop a novel peat depth model for Dartmoor in Southwest England.

The challenge of using airborne electromagnetics (AEM) for peat volume mapping is to resolve shallow and thin peat layers in the order of a few meters, particularly if the resistivity of the underlying stratum is close to the resistivity of peat. On the other hand, AEM allows the mapping of greater thicknesses of peat and (unlike AGR) without the need for depth calibrations, except for ground truthing purposes. Examples for AEM peat surveys are rare. Puranen et al. [11] and Airo et al. [7] published results from Finland, and Beamish and Farr [12] from Wales, using fixed-wing frequency-domain systems. More recently, Silvestri et al. [13,14] presented peat thickness results acquired by helicopter-borne time-domain electromagnetic devices in Norway and Indonesia.

Here, we propose a combination of both methods using helicopter-borne radiometric and frequency-domain electromagnetic systems for the large-scale (>10 km$^2$) mapping of peat volumes. We demonstrate the feasibility by a case study at a bog in Germany.

## 2. Materials and Methods

### 2.1. Study Area: Ahlen-Falkenberger Moor

The Atlantic peat bog complex, the Ahlen-Falkenberger Moor (AFM), is situated about 20 km to the south of the Elbe River estuary in Northwest Germany (Figure 1). It is located in a Pleistocene depression at the border between the coastal marshland (Holocene clay) and Saalian deposits (Pleistocene sand partly mixed with gravel and loam). This situation is reflected within the peat profiles [15]. Whereas the northern and northeastern peat profiles in the peat bog complex are characterized by marine clay deposits, the central and southern peat profiles are lacking these deposits (Figures 2 and 3). Paludification caused fen peat formation approximately 4000–5000 years B.C., followed by bog formation either on the underlying fen peat or directly on the Pleistocene sand (Figure 3). The latter mainly developed around the sand ridges, breaking through the peat bog complex near the village Ahlen-Falkenberg. In general, the stratigraphy of the peat bog complex consists of fen peat at the bottom of the profile, covered by hemic to sapric bog peat and fibric bog peat at the top, each 1–2 m thick [16].

Since the Middle Ages the AFM has either been used for small scale peat cutting at the edges or for industrial peat cutting (since 1957) within the peat bog complex. After the industrial peat cutting was terminated in 2002, these areas were re-wetted for nature conservation purposes. After the cultivation of the peat bog complex at the beginning of the 20th century, associated with an establishment of 50 homesteads, the AFM has been mainly used as grassland [17]. Only a small area in the southeastern part of the peat bog complex has never been drained for any purpose [18].

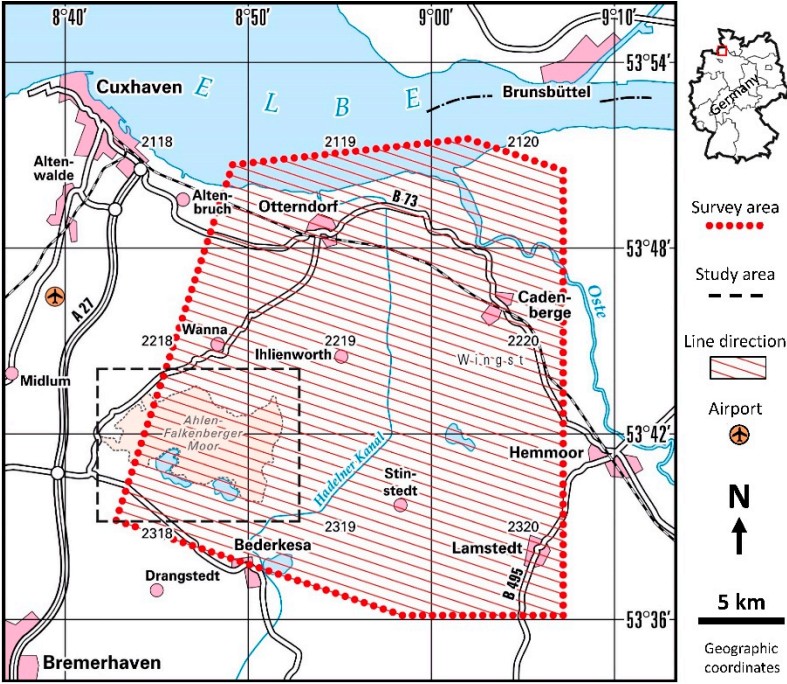

**Figure 1.** Airborne survey area (red dots) and location of the study area (dashed black rectangle) of the Ahlen-Falkenberger Moor in northwest Germany.

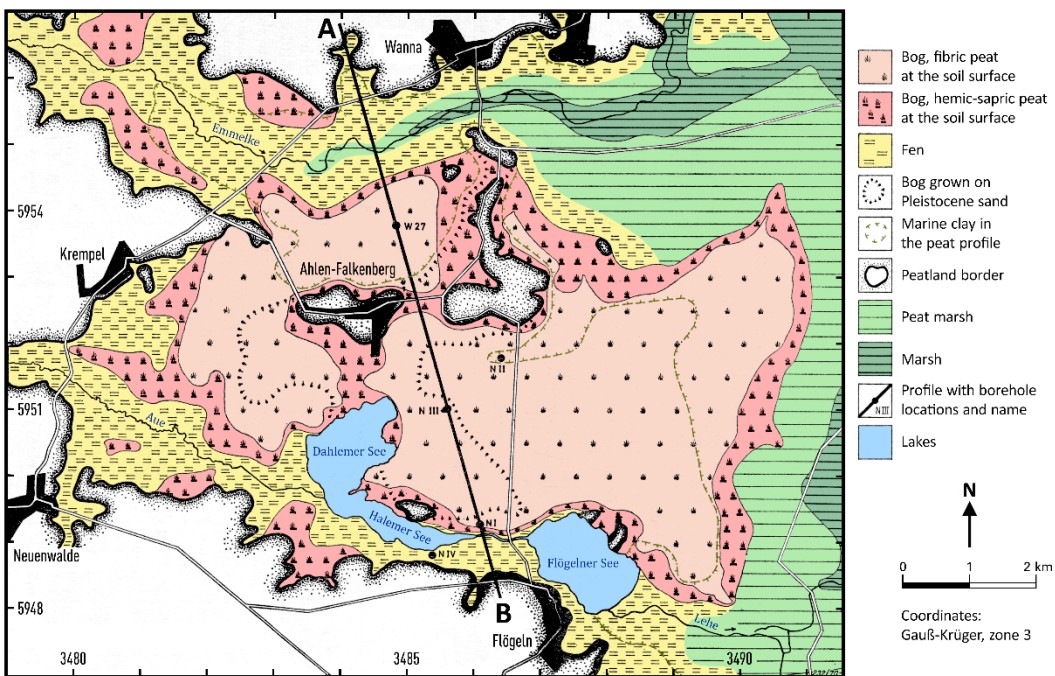

**Figure 2.** Quaternary geology map of the study area (after [15], colors and legend changed).

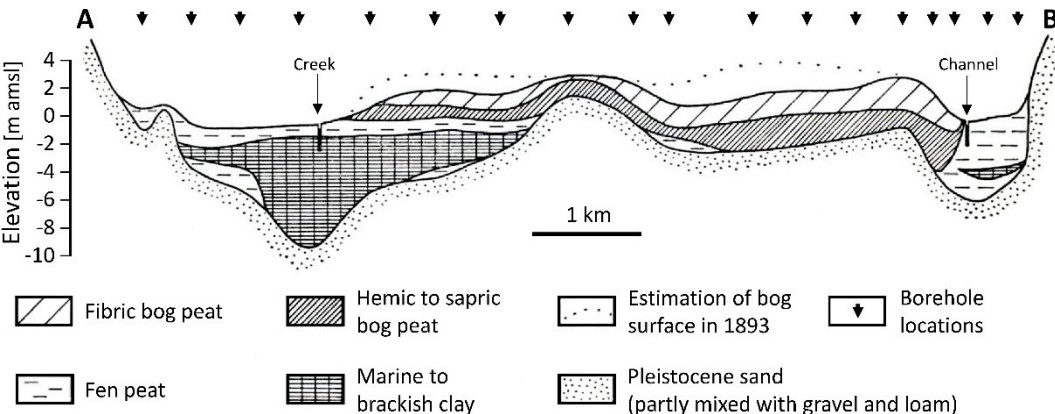

**Figure 3.** Geological profile A–B of Figure 2 (after [15], legend changed).

The surface elevation of the bog ranges from a few decimeters to several meters above mean sea level (amsl). Figure 4 shows a map derived from a digital elevation model at 25 m cell size (DGM25 [19]).

Recent studies showed that the peat is on average $t_m = 2.9 \pm 1.1$ m thick and the peat base lies at $z_m = -0.3 \pm 1.3$ m amsl within the area of the bog. The mean values were derived from 110 boreholes with peat thicknesses of 0.4–7.0 m (Figure 4) drilled with a gouge auger (Eijkelkamp, Giesbeek, The Netherlands) by the State Authority for Mining, Energy and Geology (LBEG) in 2007 [20]. Peat profile descriptions were performed after Caspers et al. [21]. Within the peat bog complex, the ground-water level is determined by land use. In re-wetted and near natural areas the ground-water level is at the surface, and in grassland areas, where drainage channels or pipes exist, it is close to the surface (<1 m depth [18]).

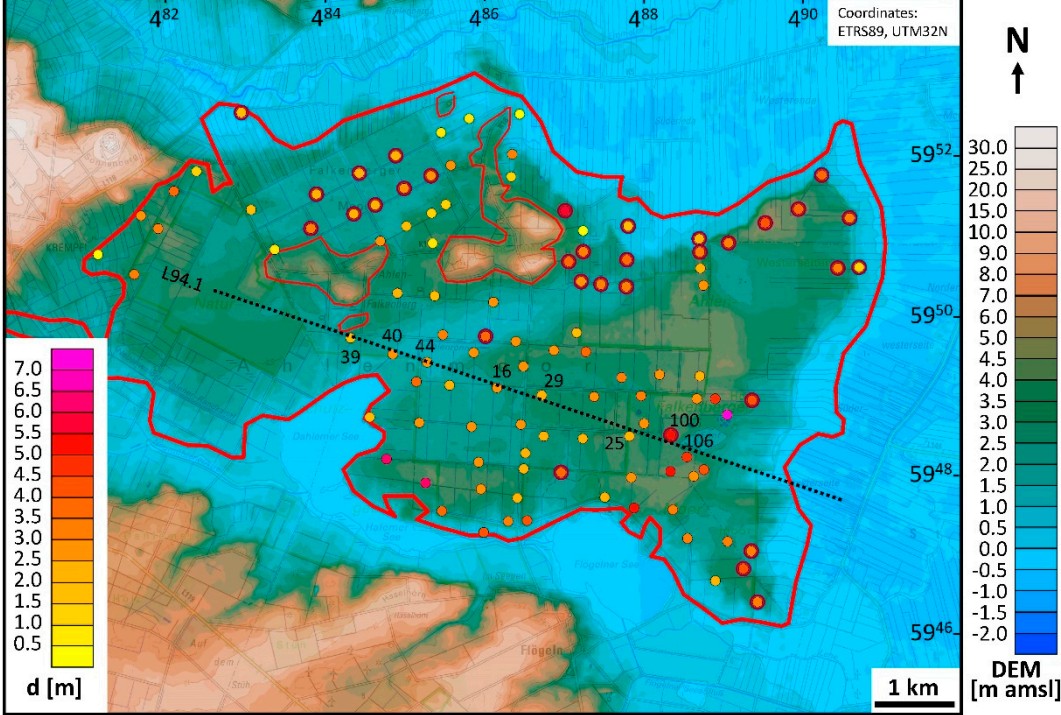

**Figure 4.** Digital elevation model DGM25 [19]. The thick red line outlines the bog and the thin red lines display the location of Geest highs inside the bog area [15]. The colored dots show the location of boreholes and indicate peat thicknesses above sandy (smaller dots) or clayey (larger red-rimmed dots) soil sediments found in 110 boreholes [20]. The dotted line indicates a flight line (L94.1) close to selected boreholes (numbers). Background: DTK50-V [22].

## 2.2. Airborne Geophysical Data

Airborne geophysics is a modern geophysical mapping technique. The Federal Institute for Geosciences and Natural Resources (BGR) operates a helicopter-borne survey system, which simultaneously records electromagnetic, magnetic, and radiometric, as well as position data (Figure 5). Helicopter-borne frequency-domain electromagnetics (HEM) is an active method, which uses transmitters (coils) to generate time-varying magnetic dipole fields (primary fields) in order to induce eddy currents in an electrically conductive subsurface. In turn, these eddy currents generate secondary magnetic fields, which are picked up by receivers (coils). The ratios of secondary and primary fields (measured in ppm) were used to reveal the subsurface conductivity distribution. Helicopter-borne radiometrics (HRD) is a passive method that measured the natural radioactivity of the shallow subsurface. A gamma-ray spectrometer detected the radiation from the three naturally occurring radioelements potassium, uranium, and thorium in the soil. Magnetic data, however, were not used in this study.

### 2.2.1. Helicopter-Borne Survey

Over the past two decades, this system has been used to map large parts of the German North Sea coastal region [23]. One of the survey areas (Hadelner Marsch) is located to the south of the estuary of the river Elbe, covering a large part of the AFM (Figure 1). The reprocessed HEM and HRD data sets and the resulting products of this airborne survey are publicly available via BGR's product center [24–27].

BGR conducted the airborne survey (about 700 km$^2$, 3000 line-km) in spring 2004 [28]. At that time, the survey system consisted of a towed bird (type: Resolve) with a digital five-frequency HEM system (f = 0.4–140 kHz) as well as a 256-channel gamma-ray spectrometer (type: GR-820) with a 4 × 4 l sodium iodide (NaI) detector pack installed in the helicopter (Figure 5). The Resolve bird also carries a magnetometer, a laser altimeter, and a GPS receiver. Most of the data sets were sampled at 10 Hz, except the radiometric data, which were sampled at 1 Hz.

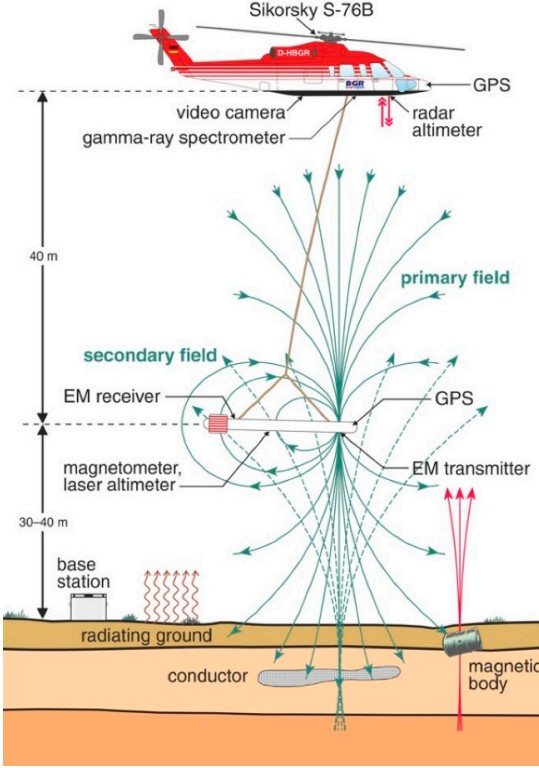

**Figure 5.** Sketch of BGR's helicopter-borne geophysical system (after [29]).

### 2.2.2. Data Processing

The aim of data processing is to generate information from the raw data that is useful for geological interpretation. This also includes the elimination of those portions in the data that are affected by influences not belonging to the subsurface. We processed the airborne data using in-house software tools [30] and Geosoft's software package Oasis montaj [31], which is suitable for the processing and visualization of airborne and other large data sets. All maps presented here were based on minimum curvature grids [32,33] of cell size 50 m, 50 m search radius, 200 m blanking distance, 100 m extrapolation, and tension 1.

In connection with another project [34], we recently reprocessed the entire data set of the airborne survey. After a literature study and our own promising preliminary results, we decided to evaluate the reprocessed electromagnetic and radiometric data in order to estimate peat thickness and extent. The magnetic data, however, appeared to be less useful for peat mapping, because the magnetic signature showed no obvious correlation with the distribution of peat in the survey area.

Position Data Processing

The Resolve bird is equipped with GPS and a laser altimeter (Figure 5) to record bird position and altitude data. Both measurements are prone to errors. While GPS data show drifts and jumps of varying magnitude, tree canopy and attitude effects affect laser data. In order to detect these errors, we compared the topography calculated from GPS elevation and laser altitude with a digital elevation model (DEM) provided by BKG [19]. Such a comparison made it possible to identify and, to a certain degree, also to correct the errors to less than about ±1 m. It should also be kept in mind that the survey was conducted eight years earlier than the DEM was published (2012). In the meantime, elevation changes may have taken place in the bog area due to draining and rewetting activities [18]. This also introduced some uncertainty in the altitude determination.

HEM Data Processing

The HEM data processing included calibration, baseline leveling, correction of man-made effects, and 1D inversion. Calibration compares measured data with known quantities. Initial calibration factors provided by the manufacturer were adjusted using flights over the highly conductive North Sea water and repeated flights over an onshore test line. Baseline leveling was necessary, because temperature variations affect the system electronics. Level errors were detected and corrected using high-altitude data (baseline picking) and spatial filters (baseline adjustment) [35]. As the HEM system was not only sensitive to the subsurface conductivity distribution, but also to metallic installations at the surface, these man-made effects were detected and removed from the data [36].

The complex amplitudes of the HEM data depend on system frequency and altitude as well as on the physical parameters of the subsurface. Synthetic data, the secondary magnetic field Z (ppm) (in-phase I and quadrature Q) at an altitude h (m) above the surface and at a frequency f (Hz), were calculated for a horizontal coil pair with a coil separation r (m) by ([37]):

$$Z = I + iQ = r^3 \int_0^\infty R_1(f, \lambda, \rho, \mu, \varepsilon) \frac{\lambda^3 e^{-2\alpha_0 h}}{\alpha_0} J_0(\lambda r) d\lambda, \tag{1}$$

where $\alpha_0 = \lambda^2 - \omega^2 \mu_0 \varepsilon_0 + i\omega\mu_0/\rho_0$ with $\lambda$ = wave number, $\omega = 2\pi f$, i = imaginary unit, $\mu_0 = 4\pi \times 10^{-7}$ Vs/Am, $\varepsilon_0 = 8.854 \times 10^{-12}$ As/Vm and $\rho_0 > 10^8$ Ωm. $J_0$ is the Bessel function of first kind and zero order. $R_1$ is the complex reflection factor containing the isotropic material parameters resistivity $\rho$ (Ωm) (reciprocal of the electrical conductivity $\sigma$ (S/m)), dielectric permittivity $\varepsilon$ (As/Vm), and magnetic permeability $\mu$ (Vs/Am). The latter two material parameters were set to their values in air ($\varepsilon_0$, $\mu_0$).

In order to reveal the subsurface parameters, the HEM data have to be inverted. Inversion tries to find a model that explains the data. Therefore, synthetic HEM data belonging to a specific model are

compared with measured data and corrections are iteratively derived from the differences. Generally, the (measured) secondary field data Z are inverted to resistivity (ignoring the often weak influence of dielectric permittivity and magnetic permeability) using two principal models, the homogeneous half-space and the layered half-space.

The homogeneous half-space inversion uses single-frequency data. The two data values, I and Q, are transformed to two model parameters, apparent resistivity $\rho_a$ (half-space resistivity) and apparent distance $D_a$ of the HEM sensor to the top of the half-space (air-layer thickness). These values are picked from once calculated polynomials, look-up tables, or grids [38]. If the subsurface is not homogeneous, calculated and measured air-layer thickness may differ. Their difference, the apparent depth $d_a$:

$$d_a = D_a - h, \tag{2}$$

reveals information on a layering of the subsurface. Positive/negative values occur, if—with respect to the half-space resistivity—a more resistive/conductive cover layer exists. Therefore, the apparent resistivity is an estimate for the half-space resistivity below a cover layer (if existent). The centroid depth z* [38]:

$$z^* = d_a + 251\,(\rho_a/f)^{1/2}, \tag{3}$$

is regarded as the center of current flow in a homogeneous half-space [39]. It is therefore a measure for the mean depth of investigation at a certain frequency. The centroid depth is proportional to the square root of the apparent resistivity, corrected by the apparent depth.

Multi-layer (or one-dimensional, 1D) inversion is able to take the data of all $N_f$ frequencies available into account. The result is an M-layered model consisting of M resistivities and M − 1 thicknesses. This kind of model is particularly important to reveal the dominant layers or interfaces of a subsurface. We applied a Levenberg–Marquardt single-site inversion approach, i.e., without lateral constraints, using starting models derived from $\rho_a - z^*$ sounding curves [40,41]. Smooth models are achieved by increasing the number of layers and the strength of regularization [42]. We applied a smoothness factor of 2.8. As the HEM inversion in this study focused on the shallower portion of the subsurface resolvable by HEM, the starting models consisted of many thin layers with gradually increasing thicknesses [43]. The layer thicknesses accumulated to a maximum model depth that was halved compared to the normally used value. All thicknesses were fixed during the iterative inversion procedure except that of the cover layer, which started at 0.5 m. The iterative inversion procedure stopped at a 10% gradual decrease of the misfit q:

$$q[\%] = \frac{100}{N} \sum_{j=1}^{N} \frac{\left| Z_j^m - Z_j^d \right|}{Z_j^d}, \tag{4}$$

where $Z_j^m$ and $Z_j^d$ are in-phase and quadrature components of the synthetic or measured (d) and modelled (m) data and $N = 2\,N_f$.

HRD Data Processing

The spectra recorded by the gamma-ray spectrometer were processed, largely using standard procedures described in the recommendations of the International Atomic Energy Agency (IAEA) [44]. After a gain stabilization, the so-called window method was applied to the spectra. This method is based on energy windows within the gamma spectrum that are associated with the three naturally occurring radioelements potassium (K), uranium (U), and thorium (Th). The windows are located around the 1460 keV peak of K-40, the most prominent gamma peak in the U-238 decay chain (1765 keV of Bi-214) and the most prominent peak in the Th-232 decay chain (2614 keV of Tl-208). The integral count rate within each window is then related to the abundance of the associated radioelement, assuming that the decay products Bi-214 and Tl-208 are in equilibrium with their parent elements. Various correction and

processing steps are applied to the window count rates using system specific calibration parameters. These include background, stripping, and height attenuation corrections, low-pass filtering, and the application of sensitivity factors leading to (in the case of U and Th, equivalent) ground concentrations of the three elements, as well as the calculation of the exposure rate E:

$$E\ (\mu R/h) = 1.505\ K\ (\%) + 0.653\ eU\ (ppm) + 0.287\ eTh\ (ppm). \tag{5}$$

The final data set was corrected for anthropogenic effects (shielding effect of infrastructure) by removal and interpolation of the affected data sections. A filter was applied to the final data in the grid domain in order to lower the high noise portion, which is a result of the statistical properties of radioactive decay.

### 2.2.3. Peat Volume Estimation

The first step of peat volume estimation (Figure 6) was to outline the bog area. This was achieved using low airborne gamma-ray data (HRD) and elevated surface data (DEM). For both, site-dependent thresholds were necessary to define a peat indicator. The next step was to analyze the airborne electromagnetic (HEM) data in order to derive estimates for the location of the peat base. This interface should be detectable, if the resistivity of peat differs significantly from the resistivity of the substrate [11–14]. The resulting peat thickness could then be used to scale the HRD data (exposure rate). Scaling was required because lowered radiation indirectly yields peat thickness values [2,10]. Finally, the peat volume of the bog was estimated from averaged HEM and HRD peat thicknesses and the area of the bog. Boreholes were only used to evaluate the airborne results.

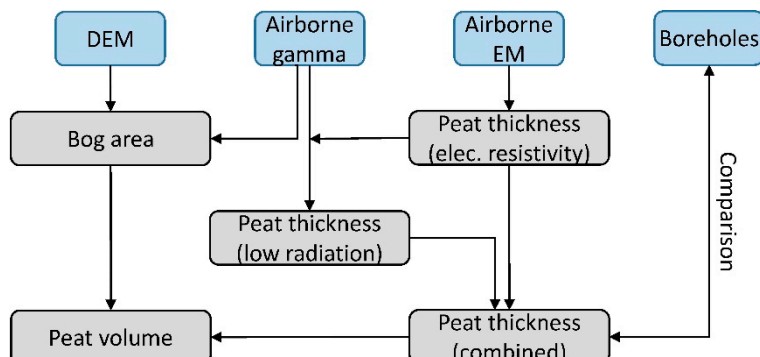

**Figure 6.** Workflow (blue colored: data, grey colored: objectives) for the estimation of peat volumes.

### HEM Analysis

In order to reveal the peat base from HEM data, we calculated smooth inversion models using all available HEM data in combination with a steepest gradient (SG) approach. This SG approach was developed to derive fresh–saline groundwater interfaces, i.e., the boundary between a poor conductor above a good conductor, from smooth resistivity models [42]. Here, the opposite case existed: a better conductor (wet peat) above a poor conductor (predominately freshwater-saturated sand). This case was more challenging, because the gradients, which we derived from the inversion models after spline interpolation [45], were less steep. Furthermore, the peat base was often very shallow and, thus, difficult to resolve by HEM. We demonstrated this by the following modeling study. The number of model layers, the control parameters of the inversion, and the steepest gradient analysis were identical to those used for field data.

The models chosen represent typical, but simplified situations. We regarded two three-layer cases (cover, peat, and substrate). The parameters of the thin resistive cover (vadose zone, $\rho_1 = 35\ \Omega m$, $d_1 = 0.25\ m$) and the resistive substrate (sandy half-space, $\rho_3 = 150\ \Omega m$) were kept constant and only the parameters of the second layer were varied. In model 1, the thickness of a moderately conductive ($\rho_2 = 35\ \Omega m$) peat layer was increased ($d_2 = 0.25$–$10\ m$). In model 2, the resistivity of the second

layer of constant thickness ($d_2$ = 3 m) was changed ($\rho_2$ = 1–200 $\Omega$m). The model parameters used were adapted from field results and literature values [11,12]. In order to simulate realistic field data, 1% random noise was added to the synthetic data.

Results of inversion and steepest gradient analysis are shown in Figure 7. A general result is that the majority of the steepest gradients, which should represent the interface between the variable second layer and the half-space, were close to the model values, although noisy data were used. Greater deviations occurred, as expected, if the layer thickness was too small ($d_2 < 1$ m in model 1, Figure 7a) or the resistivity too close to that of the half-space (above $\rho_2 = 100$ $\Omega$m in model 2, Figure 7b). The deviation at about $d_2 = 4$–7 m layer thickness in model 1, however, was somewhat surprising. There, the transition from conductive to more resistive layers was very smooth in the inversion models, so that the indication of a clear steepest gradient was difficult. The reason for this might be the degree of smoothness chosen for the inversion, which was a compromise between achieving maximal resolution and avoiding oscillations. Despite this challenge, the modeling study demonstrated that it should be possible to reveal peat thicknesses from smooth HEM inversion results. It is worth noting that steepest gradients also occurred for very low second-layer resistivities (below $\rho_2 = 10$ $\Omega$m in model 2). These resistivities are typical for clayey sediments saturated with fresh, brackish, or saline water. Therefore, additional resistivity thresholds (lower and upper bound) and a non-linear filter [46] should be applied to select appropriate steepest gradient values for a peat-base estimation.

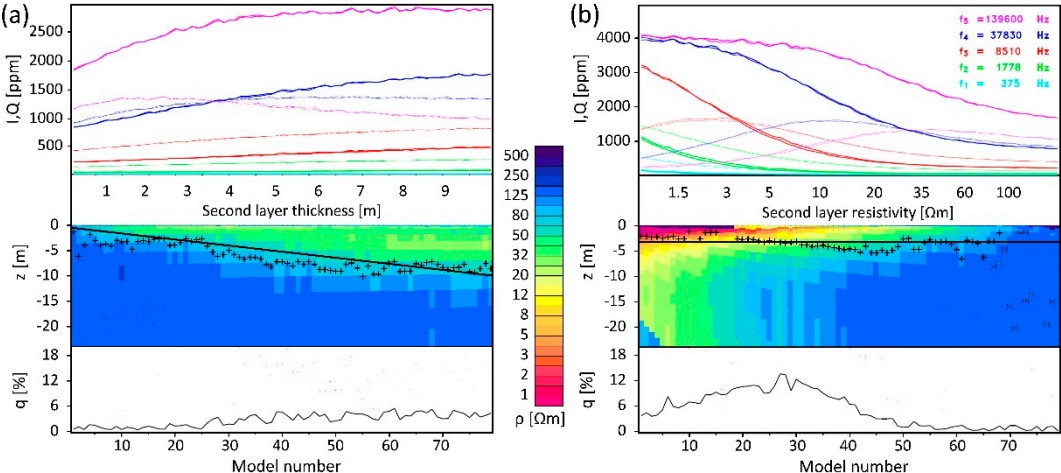

**Figure 7.** Results of a modeling study showing, from top to bottom: noisy five-frequency HEM data (I and Q before and after inversion), 1D inversion models (colors), and steepest gradient analysis (+), as well as the misfit q of the inversion. Models: Cover layer ($\rho_1 = 200$ $\Omega$m, $d_1 = 0.25$ m), variable layer (**a**) model 1 ($\rho_2 = 35$ $\Omega$m, $d_2 = 0.25$–10 m) and (**b**) model 2 ($\rho_2 = 1$–200 $\Omega$m, $d_2 = 3$ m), underlying half-space ($\rho_3 = 150$ $\Omega$m). The thick black lines indicate the top of the half-spaces.

HRD Analysis

A common approach to interpret gamma-ray data is to assume exponential absorption of the radiation passing through a homogenous material of thickness d:

$$I = I_0\, e^{-\mu d}, \tag{6}$$

where I and $I_0$ are the observed and initial radiation intensity [47]. The linear attenuation coefficient $\mu$ of the material is an intrinsic property of each material. In order to simulate the expected radiation over peatlands, a three-phase soil had to be regarded to calculate the effective attenuation coefficient:

$$\mu_e = \mu_p P_p + \mu_w P_w + \mu_a P_a, \tag{7}$$

where $\mu_x$ and $P_x$ are the attenuation coefficients and fractions of peat (p), water (w), and air (a), respectively [48,49]. Figure 8 displays the effective linear attenuation coefficients for diverse mixtures of the three phases. Using literature values ($\mu_p = 0.528$ m$^{-1}$, $\mu_w = 5.72$ m$^{-1}$, and $\mu_a = 0.068$ m$^{-1}$, e.g., [2]) and assuming realistic conditions for peatlands (80% porosity and 80% water saturation), $\mu_e$ should be close to 3.8.

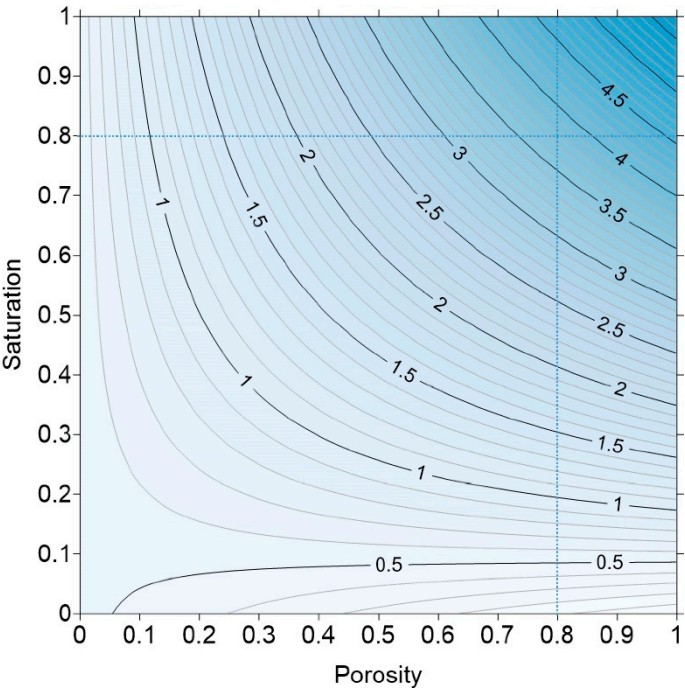

**Figure 8.** Effective linear attenuation coefficient $\mu_e$ as a function of porosity and water saturation. The dotted blue lines indicate the 80% values.

Figure 9 shows the theoretical attenuation $I/I_0$ with respect to peat thickness (log-lin scale) for diverse scenarios, varying the porosity and the water saturation. In addition to the simple exponential approach, which is suitable for sources of small lateral extent, the theoretical attenuation generated by infinite surface sources of homogeneously distributed radioelements is shown. This calculation was based on the E2 function (exponential integral of the second kind [50,51]). The diagrams show that a peat thickness of about 1 m (80% porosity and 80% water saturation) suffices to reduce about 99% of the original radiation (from below the peat). Furthermore, unrealistically low effective attenuation coefficients requiring very low water saturation would be necessary to resolve peat thicknesses of more than 1 m.

On the other hand, Keaney et al. [10] and Gatis et al. [2] showed promising results using radiometric data for peat thickness estimation—even for peat thicknesses above 1 m. Therefore, instead of trying to model the measured radiometric data using parameters that were not known explicitly for the bog considered, we searched for an empiric relationship between peat thickness and measured radiation. We defined an apparent attenuation coefficient $\mu_a$, which was derived from a comparison of measured exposure rates with known or estimated peat thicknesses, d:

$$\mu_a = -\ln(E/E_0)/d. \qquad (8)$$

As the initial exposure rate $E_0$ of the soil beneath the peat was generally unknown, we assumed it was constant ($E_0 = 1$ µR/h).

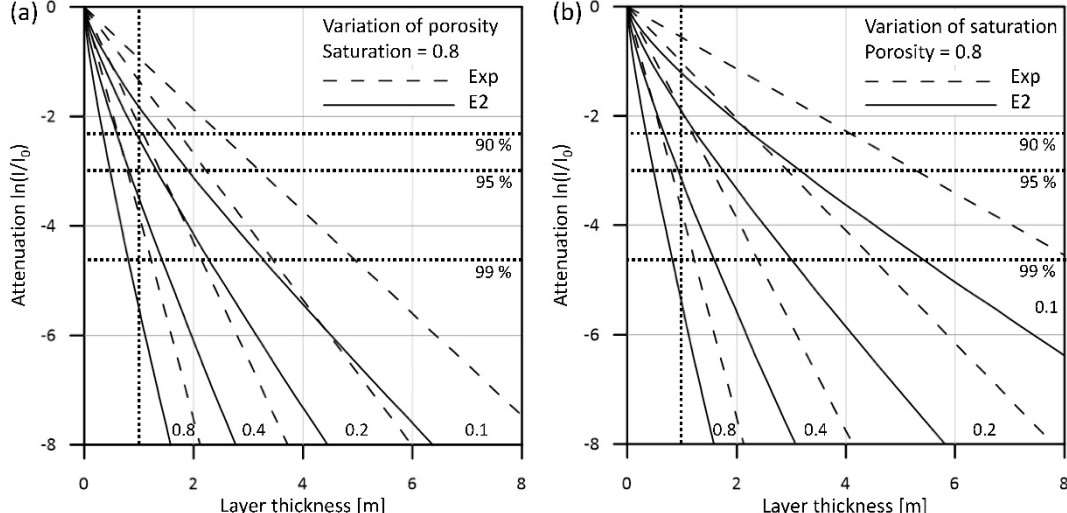

**Figure 9.** Attenuation $I/I_0$ as a function of peat layer thickness (log-lin scale) calculated for (**a**) various porosities, and (**b**) water saturations using exponential and E2 functions. The vertical dotted black line indicates the peat thickness of 1 m and the horizontal black lines indicate the 90%, 95%, and 99% attenuation.

Combined HEM and HRD Analysis

Finally, we averaged the HEM and HRD results along all flight lines used, considering thickness values with differences less than a threshold. This threshold was a compromise between using congruent results only (causing large gaps) and maximal data coverage (including averaging of inconsistent data). The resulting gaps can be interpolated based on grid values.

## 3. Results

We used the reprocessed airborne data [30] to evaluate HEM and HRD data with respect to the investigation of the AFM (Figure 2). We selected the western portion of 26 west-northwest–east-southeast flight lines (L78.1–L103.1), which crossed the AFM at 250 m line separation, for further analyses. The resulting databases are attached as Supplementary Materials.

We started with a presentation of HEM and HRD data along a selected flight line (for location see Figure 4). Figure 10 shows an HEM data example. The main variation in the HEM data (in-phase I and quadrature Q) was caused by the altitude, h, of the sensor (see Equation (1)), which generally lay between 30 and 50 m above ground level. Therefore, we transformed the HEM data to half-space parameters [38], apparent resistivity $\rho_a$ and apparent depth $d_a$ (Equation (2)), which were hardly affected by the sensor altitude. Both parameters indicated a conductive cover: the resistivity at the highest frequency was often lower than the following ones and the apparent depth was negative.

As the centroid depth of the electromagnetic fields (Equation (3)) increased with decreasing frequency (and increasing resistivity of the subsurface), the shallow subsurface was better resolved by high-frequency data. Even at the highest frequency (f = 140 kHz), however, the arithmetic mean of all centroid depth values within the area of the bog ($z^*_m$ = 5.0 ± 1.4 m) exceeded the average of the drilled peat thickness values ($t_m$ = 2.9 ± 1.1 m). It was challenging to derive peat thickness values from single-frequency HEM data, because the larger portion of the HEM signal was generated from sediments below the peat. Further challenges occurred from rather noisy highest-frequency data and from anthropogenic effects. The latter, however, could be easily corrected by the erasure and interpolation of minor effects [36], e.g., caused by the crossing road in Figure 10, and stronger effects rarely occurred within the area of the bog.

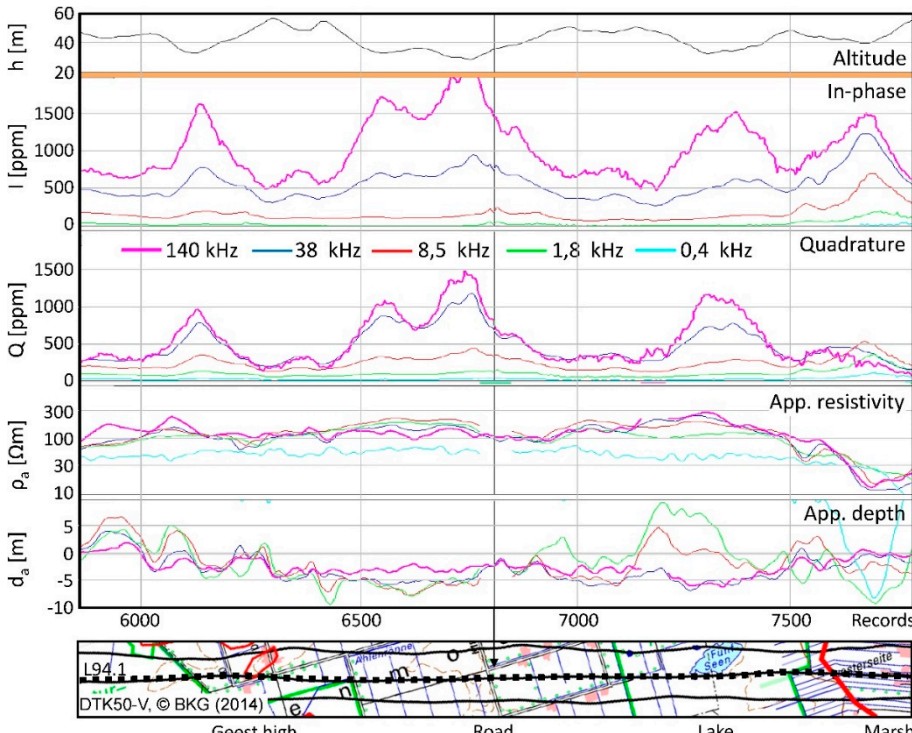

**Figure 10.** Helicopter-borne electromagnetics (HEM) data example along the western portion of flight line L94.1, showing (from top to bottom): HEM sensor altitude (h), in-phase (I), quadrature (Q), apparent resistivity ($\rho_a$), and depth ($d_a$) at five frequencies (colors), as well as an extract from a flight-line map (dotted line: L94.1). The black line indicates a road, which affected the HEM data (at lower frequencies). The short horizontal lines beneath the quadrature panel indicate erased data sections.

Figure 11 shows an HRD data example along L94.1. The high noise of the raw data demonstrates that smoothing of the final concentration (K), equivalent concentrations (eU and eTh), and total counts (TC) is necessary. The exposure rate (Equation (4)), which was the parameter used in the course of this study, was extremely low within the bog area (<1 μR/h). However, some variation appeared that may contain subsurface information. Outside the bog area (Geest high, marsh), the exposure rate clearly increased. An increase was also observed around the road, although the radiometric data close to the road (anthropogenic effect) were erased. The challenge in this study was to analyze the very low exposure rates with respect to peat thickness.

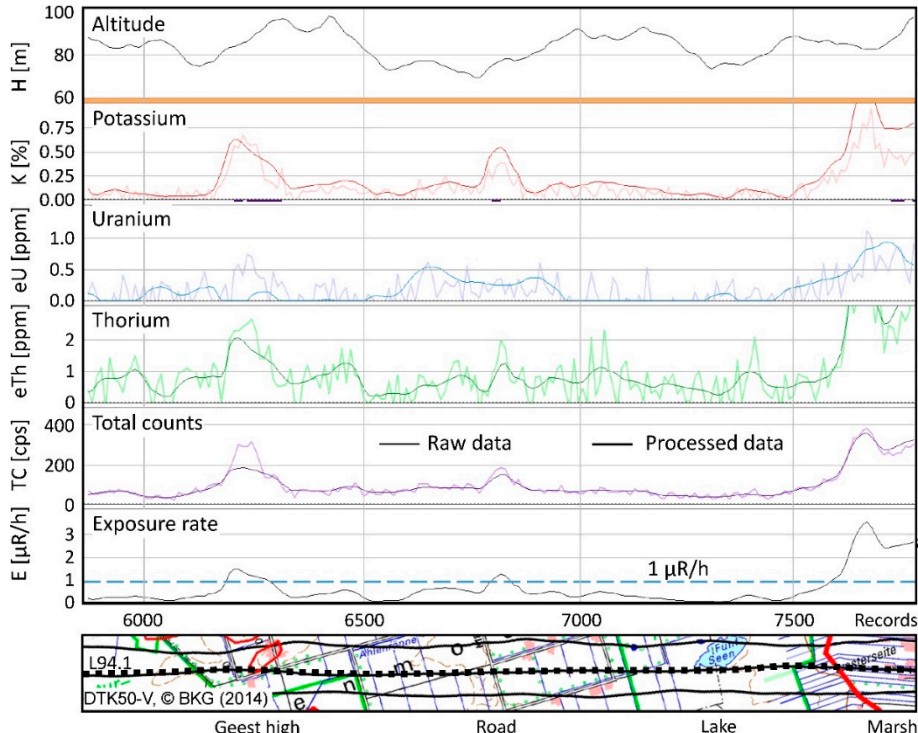

**Figure 11.** Helicopter-borne radiometrics (HRD) data example along the western portion of flight line L94.1 (see extract from a flight-line map at the bottom) showing (from top to bottom): HRD sensor altitude (H), radioelements potassium (K), uranium (U), thorium (Th), and the total counts (TC) (raw and processed data), as well as the exposure rate (E). The black horizontal lines beneath the potassium panel indicate erased and interpolated data sections.

### 3.1. Lateral Extent of the Bog

In order to find indications for the lateral extent of the bog, we inspected our standard maps used to visualize HEM and HRD results.

### 3.1.1. HEM Results

Figure 12 displays a map of the apparent resistivity at the highest frequency, which should be best suited to outline shallow features. The corresponding centroid depths (Equation (3)) were about 4–7 m in the blue-colored areas and about 3–5 m in the yellow/green colored areas. Partly, there was a good correlation between high apparent resistivities ($\rho_a > 100$ Ωm) and the outline of the bog (thick red line). The area of the fen (which partly underlies the bog) and the marsh areas were characterized by lower apparent resistivities ($\rho_a < 50$ Ωm). The sandy Geest areas in the south (and northwest) beyond the outline of the peatland (thin brown line) were again resistive. A clear correlation between peat extent (bog) and apparent resistivity, however, was not obvious, because both high and low resistivity values appeared within the area of the bog. The green line, which outlines the distribution of marine clays in the northeastern portion of the map [15], approximately separates high and low resistivities. The existence of clayey sediments below the peat was also confirmed by boreholes (red circles), in which clay, silt, or till were found at the base. Therefore, the apparent resistivities—even at the highest frequency—related more to the sediments below the peat (indicating sandy and clayey substrate) than to the peat itself.

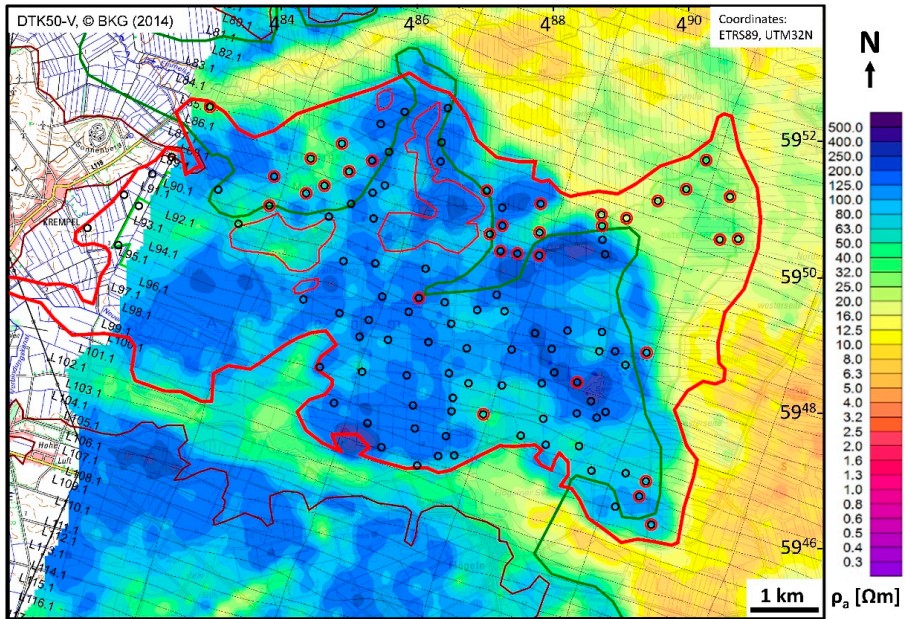

**Figure 12.** Apparent resistivity at the highest frequency (f = 140 kHz). Blue/orange colors indicate sandy/clayey substrate. Red circles mark boreholes with clay at the base. Red lines outline the bog, brown lines indicate the boundary of the peatland, and the green line displays the southernmost extent of marine clays [15]. The dotted lines are the numbered flight lines.

### 3.1.2. HRD Results

The outlines of the bog (Figure 13, red lines) were fairly correlated with extremely low exposure rates (E < 1 μR/h). Outside the bog (and fen), where sandy Geest and clayey marsh sediments predominate, higher exposure rates occurred. Only in the south, where the surface elevation was lower, there were similar low exposure rates caused by lakes, fens, and smaller bogs. Therefore, lowered radiation (Figure 13) alone cannot sufficiently indicate the bog area.

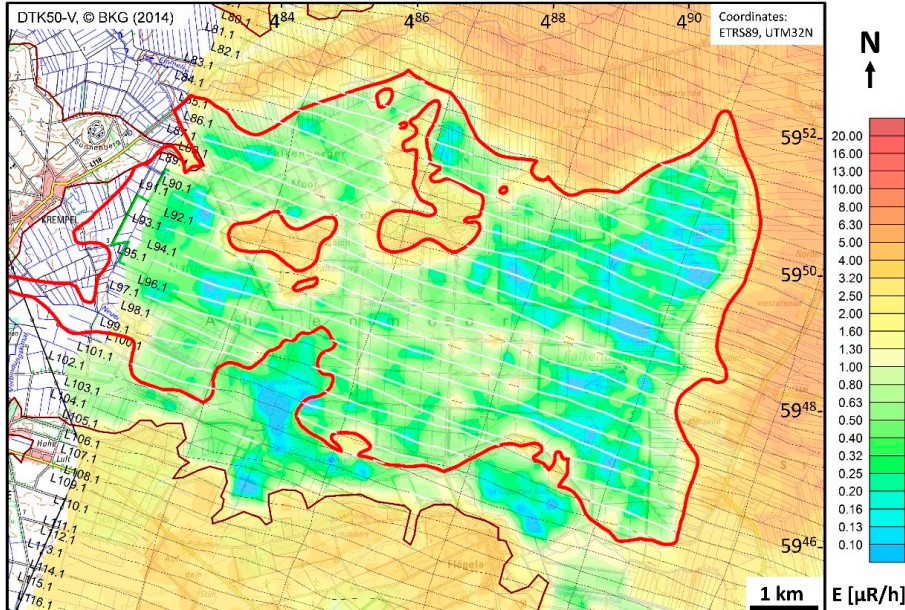

**Figure 13.** Low exposure rates (green to blue colors) correlate fairly well with the extent of the bog (red lines), except in the south, where lakes, fens, and smaller bogs exist within the area of the peatland (brown lines) [15]. The white lines on the black flight lines represent the peat indicator derived from areas of low radiation and higher elevation.

A further indicator for a raised bog is its surface elevation (Figure 4). This elevation, however, was also high outside the bog, e.g., in the Geest area. The combination of both helped to outline the bog area. Using two thresholds, E < 1 μR/h (radiation) and DEM > 0.5 m amsl (elevation), we defined a peat indicator for the bog extent (white lines in Figure 13). We used this peat indicator to select and display the results.

### 3.2. Vertical Extent of the Peat Within the Area of the Bog

#### 3.2.1. HEM Results

The results of Figure 12 demonstrate that single-frequency apparent resistivities derived from HEM data, even at the most suitable frequency, did not suffice to reveal the lateral peat extent. As it was also not possible to derive layer thicknesses from single-frequency data, we estimated peat thicknesses from smooth 20-layer HEM inversion models using the entire five-frequency HEM data set. Some of the results derived from the inversion of field data are shown in Figure 14 (results close to eight boreholes) and Figure 15 (flight line L94.1). There, the cover-layer thickness was often reduced (i.e., <0.5 m) in the inversion models and the resistivity flipped between lower ($\rho_1 < 30$ Ωm) and higher ($\rho_1 > 200$ Ωm) values. This cover-layer may indicate upper soil conditions (wet or dry), but remaining inaccuracies of laser altitude measurements may also affect the inverted cover-layer parameters. Nevertheless, the resulting HEM models showed an increase in resistivity close to the drilled peat base (PB). We estimated the depth of the peat base with the help of the steepest positive log(ρ)–log(z) gradients (SG) derived from the spline interpolated HEM models. Sometimes, however, larger deviations did occur, e.g., at borehole 39 (for location see Figure 4), which is located close to the border of the bog near a Geest high.

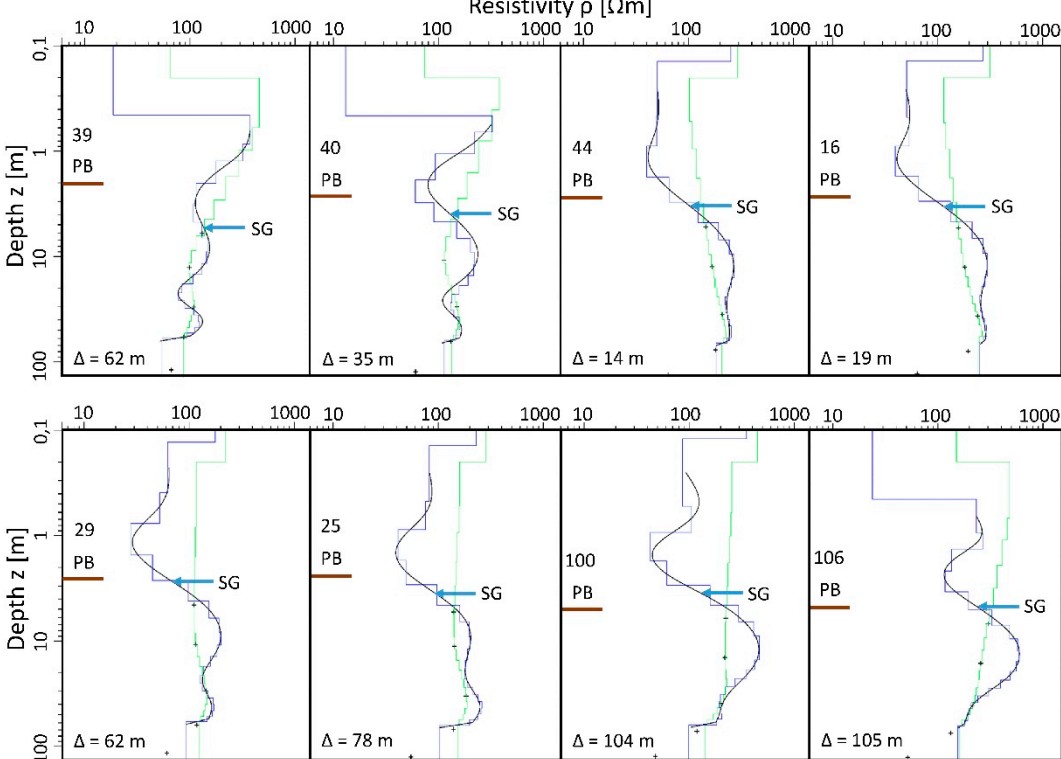

**Figure 14.** Comparison of peat thicknesses derived from steepest gradient (SG, blue arrows) analysis of smooth 20-layer HEM inversion models (green/blue/black: starting/final/interpolated models) and drilled peat base (PB, brown lines) along flight line L94.1. The distances of the boreholes (numbers) from the flight line (Δ) are indicated.

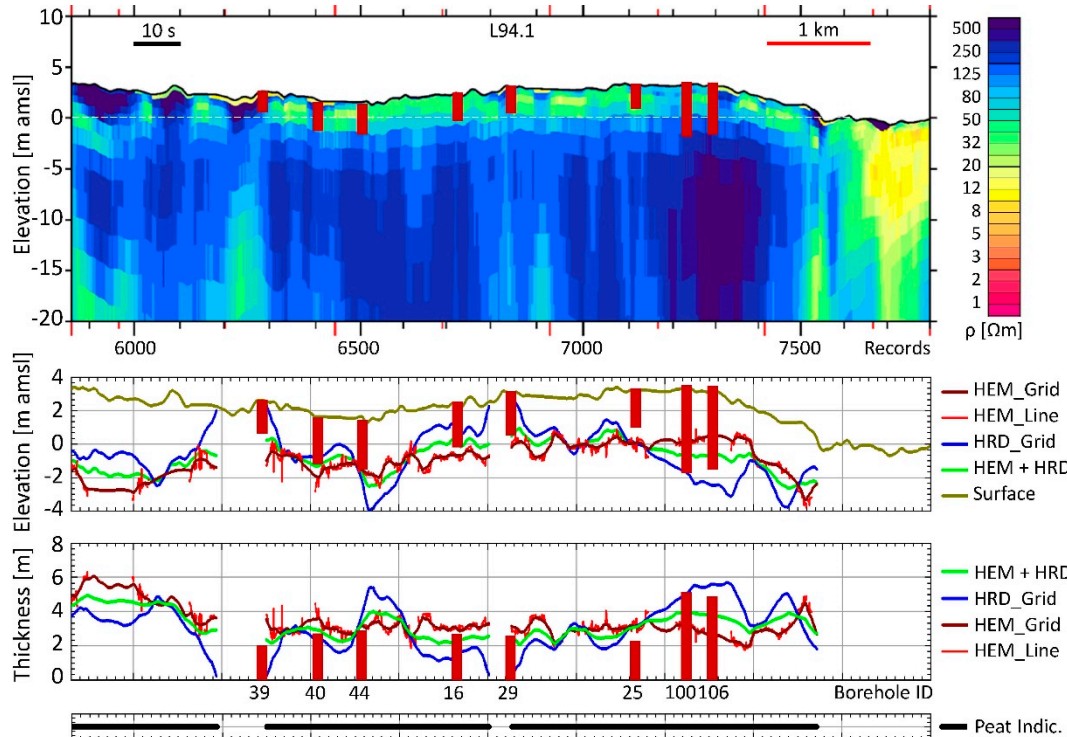

**Figure 15.** Vertical resistivity section along flight line L94.1 (above) derived from smooth inversion of HEM data, and peat elevations/thicknesses (below) derived from HEM or/and HRD data in comparison with peat ranges found in boreholes (brown columns). HEM_line: SG analysis along lines, HEM_Grid: gridded HEM results, HRD_Grid: gridded HRD results, HEM + HRD: combined results. Surface elevation and a peat indicator (derived from airborne data) are also shown.

In order to exclude unrealistic interface estimates derived from SG values along the flight lines, e.g., along flight line L94.1 (Figure 15), non-linear filter (length: 200 m, tolerance: 2 m) as well as resistivity ($\rho < 130$ $\Omega$m at SG interface) and elevation thresholds (above $-5$ m amsl) were used. The resulting gaps (within the red HEM_Line in Figure 15) were interpolated (dark red lines: HEM_Grid) by minimum curvature gridding (50 m cell size, 200 m search radius, 200 m blanking distance, 250 m extrapolation, maximal tension) of the interface elevation values of all 26 flight lines. We applied the non-linear filter and the interpolation to SG elevations instead of thickness values, assuming that the ancient land surface, which served as a base for the raised peat complex, was smoother than the current surface. Furthermore, the surface elevation of the bog may change with time due to land use and climate. Finally, we calculated the thickness values from the differences of surface and interface elevations. All results are displayed with respect to the peat indicator.

The resulting mean difference of the estimated interface elevations to the drilled borehole results was $\Delta_m = -0.1 \pm 1.4$ m (Figure 16), i.e., the HEM-derived peat base at 97 borehole locations selected by the peat indicator was somewhat deeper (thickness higher). One reason for this small discrepancy may be that clay, silt, or till layers below the peat (found in 1/3 of the boreholes, Figure 12) could not be clearly distinguished from peat by smooth HEM inversion. The geometric mean of the minimum resistivity values representing peat above the interface was about $\rho_m = 35$ $\Omega$m (arithmetic mean of log10 values: $1.54 \pm 0.22$), and, thus, close to the resistivity of clayey sediments, whereas the mean value below the interface representing sandy sediments was clearly above 100 $\Omega$m.

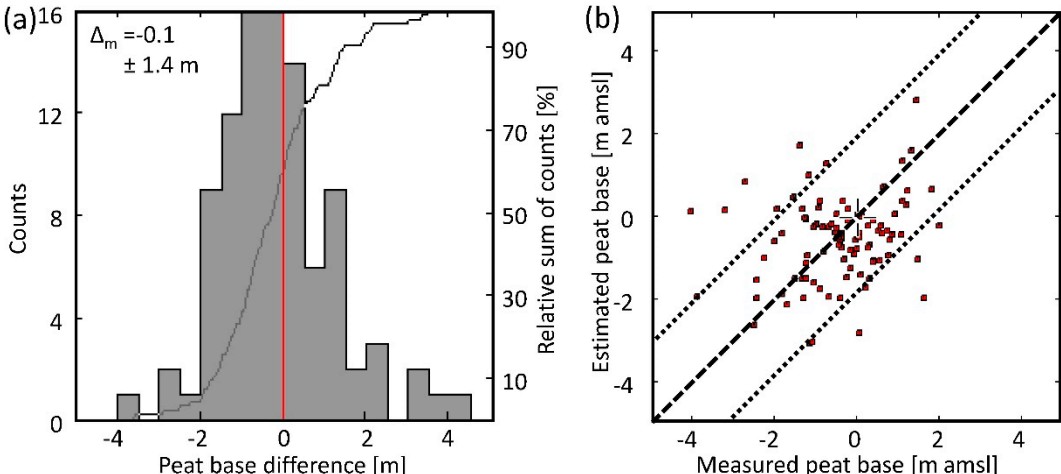

**Figure 16.** Comparison of airborne and borehole data: (**a**) histogram of differences (vertical red line: $\Delta_m = 0$ m) and (**b**) scatter plot (dashed: 1:1 line and dotted: ±2 m lines) of elevations of peat base derived from HEM.

### 3.2.2. HRD Results

We estimated the apparent attenuation coefficient $\mu_a$ (Equation (8)), assuming $E_0 = 1$ μR/h, which is close to the exposure rates at the border of the bog (Figure 13). With the help of the peat indicator, we selected the (gridded) exposure rate, E, at all borehole locations within the bog area to derive the apparent attenuation coefficients. The resulting arithmetic mean value was $\mu_a = 0.26$ m$^{-1}$ (Figure 17). The scattering was high, obviously due to undefinable effects, such as unknown initial radiation, inhomogeneity of the bog, and very low radiation. Nevertheless, there seemed to be some correlation between exposure rate and peat thickness. Using this value of $\mu_a$ for peat thickness estimation yielded a mean difference to the borehole results of $\Delta_m = -0.2 \pm 1.6$ m, which is similar to the mean difference derived from HEM, but the scattering is higher.

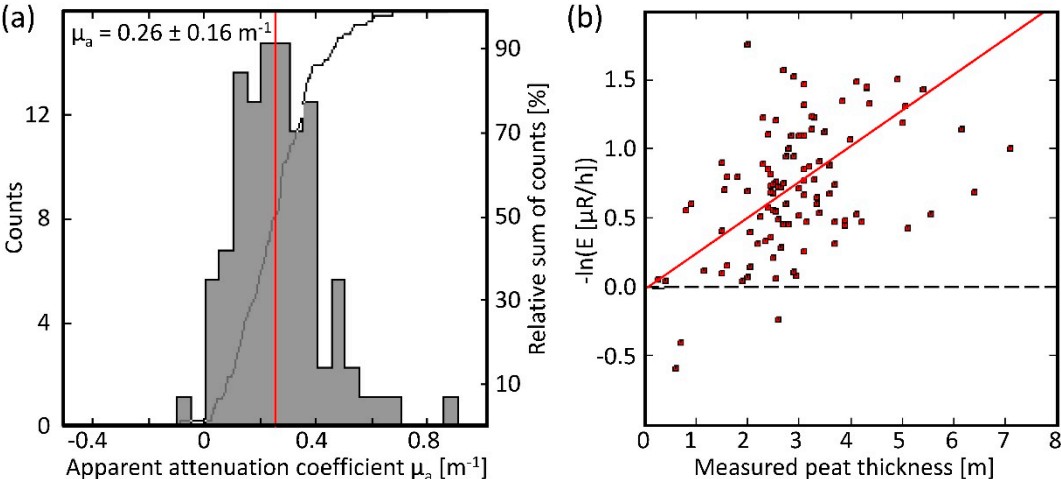

**Figure 17.** Estimation of the apparent attenuation coefficient ($\mu_a$) from exposure rates (E) and peat thicknesses found in boreholes. The red lines in the histogram (**a**) and the scatter plot (**b**) indicate the mean value and the corresponding linear relationship, assuming $E_0 = 1$ μR/h, respectively. The three negative values observed in the scatter plot belong to boreholes located close to the border of the bog.

We also tried to use other approaches (linear regression, larger $E_0$), but all these attempts resulted in larger standard deviations. Furthermore, we studied the effect of random values in order to prove that our results are significant. For this, we produced 1000 sets of 100 randomly distributed exposure rate values between 0 and 1 to calculate $\mu_a$ values using the same approach as with the real data. The resulting 1000 $\mu_a$ values showed a normal distribution with a mean value of 0.29 m$^{-1}$, which is quite near to the value of 0.26 m$^{-1}$ received from the real data. However, when we used the random $\mu_a$ values to calculate sets of peat thicknesses, their correlation to measured borehole thicknesses was significantly worse than for the set of real data. The mean of the mean difference between calculated and measured thicknesses was 0.66 m (as opposed to 0.32 m) and the mean of the standard deviation of differences was 1.98 ± 0.11 m, which is significantly larger than the standard deviation of differences resulting from the real data (1.60 m). Indeed, the latter lies well outside the three sigma limit (1.98 m − 0.33 m = 1.65 m) of the random data distribution. We therefore consider our results meaningful.

In order to estimate peat thicknesses independently from borehole data, we referred the −ln(E) values to the HEM-derived thickness values. Taking only the gridded values at the selected borehole locations into account led to mean value of $\mu_a = 0.25$ m$^{-1}$, which is close to the value received using borehole data. Thus, it should be reasonable to use the HEM-derived thickness values.

Considering all the values along the flight lines (selected by the peat indicator = 28,680 values) we derived an optimum value of $\mu_a = 0.28$ m$^{-1}$. The resulting gridded thickness values are shown in Figure 15 (HRD_Grid, blue line in the lower panel). The mean difference to the borehole results was $\Delta_m = -0.2 \pm 1.4$ m. Compared to the HEM-derived thicknesses, the HRD-derived thicknesses undulated stronger (at least on the profile shown in Figure 15), although the corresponding standard deviation was similar. Some very low thickness values occurred close to the border of the peat bog and may be explained by boundary effects (radiation from outside the peat area affected measurements inside). In addition, in areas where the measured radiation was extremely low and therefore very sensitive to the smallest leveling errors, unreasonably high thickness values were generated.

### 3.3. Combined HEM and HRD Results

We averaged (arithmetic mean) the thickness values estimated by HEM (Figure 15, brown line) and HRD (Figure 15, blue line). A threshold of ±2 m, applied to the differences, was found appropriate to exclude inconsistent results while maintaining enough data for interpolation. The resulting gaps were interpolated based on grid values. The averaged results are shown in Figure 15 (green lines) for both thickness and elevation. The thickness map shown in Figure 18 was produced with respect to the peat indicator and an expansion by two grid cells (100 m) at the edges. The total gridded area was 33 km$^2$. This peat thickness map derived from airborne data shows thicker peat occurrences (d > 4 m) in the west and east of the bog area. Unfortunately, the density of the boreholes drilled in 2007 is sparse there, but the general appearance of the airborne results coincides with a peat thickness map derived from older drilling results [15], except in the northwest. There, however, the younger borehole results also indicate peat thicknesses less than 4 m [20].

The differences in the airborne mapping and the borehole results (Figure 19) were small on average. The mean differences were $\Delta_m = 0.0 \pm 1.1$ m (thickness) and $\Delta_m = -0.1 \pm 1.1$ m amsl (peat base elevation), but locally a few greater differences above ±2 m (dotted lines in the scatter plot) did occur.

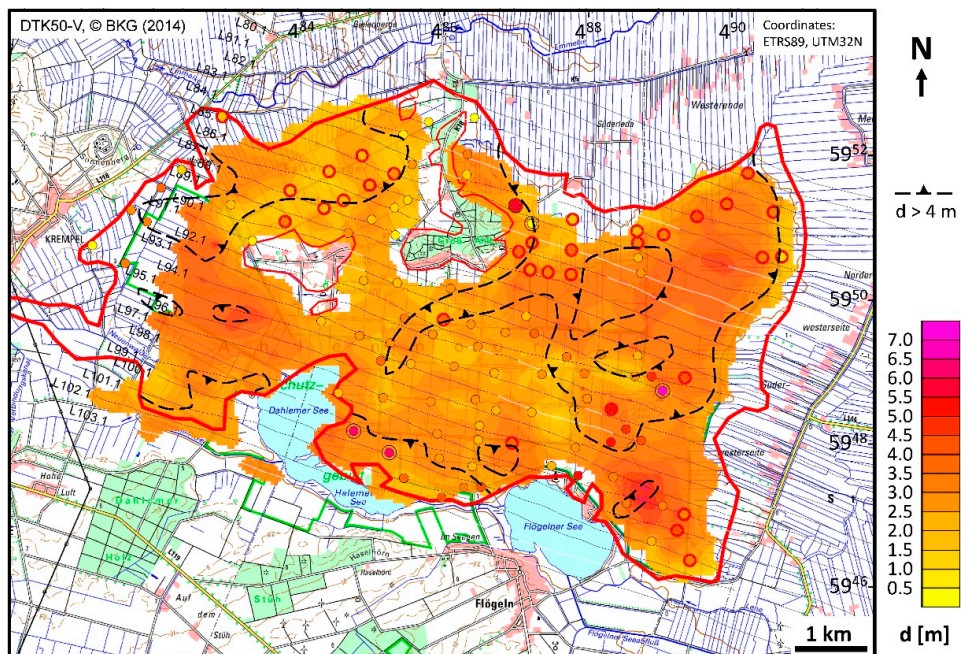

**Figure 18.** Peat thickness derived from HEM and HRD data (grid) in comparison with peat thickness found in boreholes (dots [20] and dashed lines [15]). Black/red circles indicate greater differences (above ±2 m)/clayey sediments below the peat, respectively. The black lines are the flight lines, which are colored white at places where HEM data values were interpolated at the highest frequency.

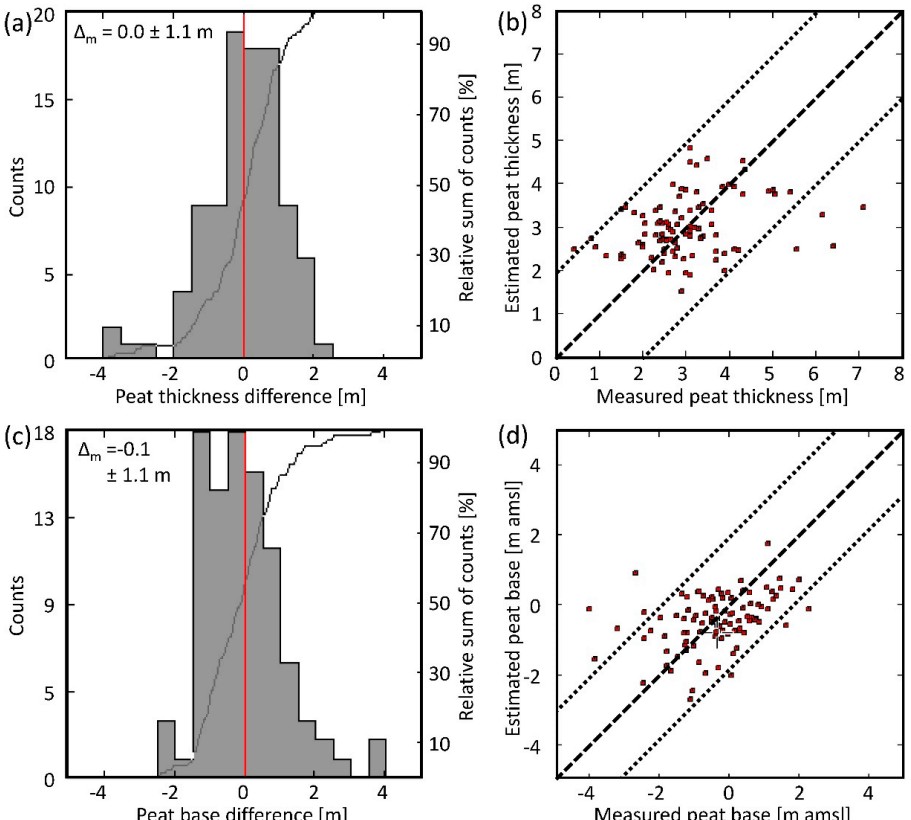

**Figure 19.** Comparison of airborne and borehole data: histograms of differences and scatter plots of peat thicknesses (**a**,**b**) and elevations of peat base (**c**,**d**) (vertical red lines: $\Delta_m = 0$ lines; dashed and dotted lines: 1:1 lines and ±2 m lines).

## 4. Discussion

The results presented above show that the estimation of plausible peat base elevations by a combination of airborne electromagnetics and radiometrics seems to be possible. However, this has been proven only at a single bog using airborne data from 2004 and borehole data from 2007. The boreholes were sparsely distributed, and in some areas, no boreholes existed at all. It is worth noting that Figure 18 compares airborne data (grids) with local borehole data (point values). For testing purposes, we applied the same gridding procedure to the borehole data. We found that their peat range reduced and some large deviations of airborne from borehole results decreased. This circumstance reflects the generally smoother appearance of airborne results compared to borehole data.

Furthermore, the boreholes used for comparison contained only limited information about the substrate (only a few decimeters), so that a discussion of the influence of the substrate on the results achieved is hardly possible. However, the grouping of results with respect to boreholes with and without clayey sediments at the base revealed that the airborne-derived peat bases are not deeper at the 31 boreholes with clay (as expected), but a few decimeters higher compared to the 66 boreholes with sand. The corresponding mean elevation differences are $\Delta_m = 0.2 \pm 0.8$ m (with clay) and $\Delta_m = -0.5 \pm 1.6$ m (with sand). A thin (few decimeters) clayey layer at the peat base may help focusing on the interface between peat and substratum (sand), but modeling studies clearly demonstrate that then the elevation of the interface is shifted downwards. As the peat above the clayey substrate mostly belongs to the fen, which widely underlies the bog (see Figure 2), it is more likely that close to the fen–clay interface a vague resistivity increase occurs. This was observed in several inversion models close to boreholes of the northeastern corner of the bog, where thick (some meters) marine silts and clays exist. A far stronger increase in resistivity from peat (fen) to underlying silt/clay appeared in the inversion models at the beginning of profile A–B (Figure 20). Thus, the steepest gradients revealed this interface successfully, particularly below the bog (flight lines 83–88).

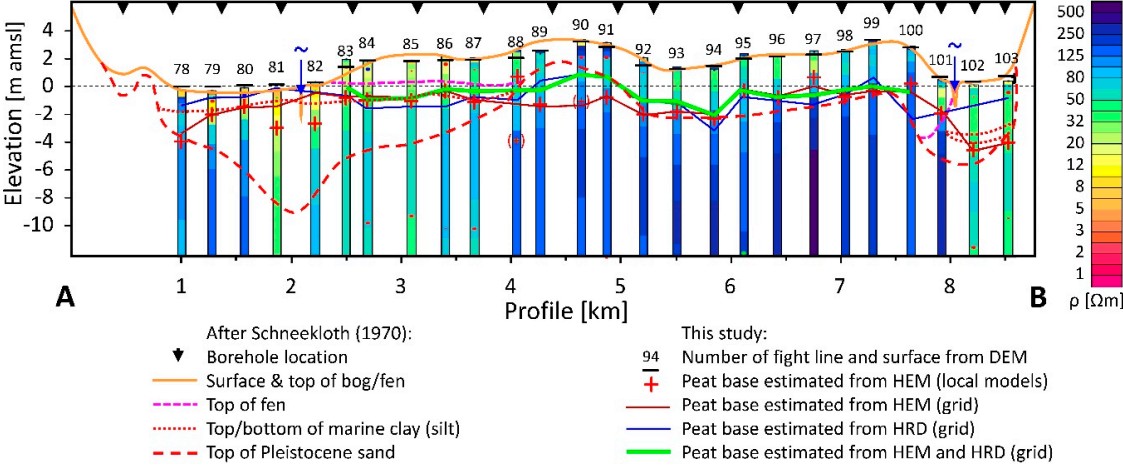

**Figure 20.** 1D HEM inversion models and comparison of peat base elevations derived by Schneekloth [16] and airborne geophysics along profile A–B (see Figure 3).

Figure 20 also demonstrates that the estimated elevation of the peat base is often close to the drilled peat base, but not always. The reasons for these discrepancies are manifold. Besides the above-mentioned differences between the footprints of the measurements and between point and grid values, there were a number of other sources of errors. The quality of data and data processing is very important, because that is the base for all following interpretation. The quality of the survey data used here is good but not excellent, particularly because the HEM system was flown for the first time (and thus not optimized), resulting in a rather high noise level on the data of the highest frequency (about 25 ppm, all others <5 ppm). This HEM noise level (1–5%) is not too critical, because filters (non-linear and low-pass filters, splines) often can reduce it sufficiently. More critical are effects on the HEM

data that are not completely known, such as remaining calibration errors, short-wavelength thermal drift, attitude, and man-made effects. Radiometric data is noisy and a number of corrections have to be applied to the data. In this context, the background correction is most critical, since it defines the zero radiation level and level shifts are easily introduced by imperfect background estimations. As described above, all these effects could be reduced to a minimum by sophisticated processing tools and calibration flights over seawater.

Another source of error was the calculation of the surface elevation from airborne measurement. Differences between this airborne-derived surface elevation and DEM values occurred. Figure 20 shows examples for such offsets at flight lines 83, 88, and 92, where the calculated elevations were about half a meter too high with respect to the DEM used [19]. The mean difference of airborne-derived surface values and the DEM along all flight lines within the area of the bog (28,747 values selected by the peat indicator) was $\Delta_m = -0.10 \pm 0.52$ m. A similar mean difference occurred comparing the airborne-derived surface values with the elevation of the 103 boreholes: $\Delta_m = -0.05 \pm 0.59$ m.

Finally, yet importantly, the peat bog complex is not homogeneous and, thus, local effects may impair the estimated peat base. On the other hand, these deviations may be used in future evaluations to reveal further information about peatlands.

One result of an airborne peatland survey could be the estimation of the total peat volume. Here, the estimated total peat volume (accumulated grid values displayed in Figure 18) was $106 \times 10^6$ m$^3$. Schneekloth [15] estimated the peat volume of the bog (fibric and hemic to sapric peat) to be more than $85 \times 10^6$ m$^3$. Taking into account that the bog is underlain largely by peat belonging to the fen (Figures 2 and 3), the airborne-derived volume estimate seems to be plausible.

## 5. Conclusions

An Atlantic peat bog complex situated in northwestern Germany was investigated in order to ascertain to what extent airborne geophysics is useful for peatland mapping. The results derived from helicopter-borne electromagnetic and radiometric data are promising. While the radiometric data in combination with surface elevation data provided the lateral extent of the bog, the electromagnetic data—even at the highest frequency of 140 kHz—revealed the substrate of the bog, which is predominantly sand, and partly clay, silt, and till.

The electromagnetic data, after inversion to smooth 20-layer resistivity-depth models, also provided estimates of the elevation of the peat base, which are on average only a decimeter below the peat base derived from boreholes. The layer with clayey sediments at the base of the peat did not lower the elevation of peat base estimates. On average, the water-saturated peat seems to be somewhat more resistive (about 35 $\Omega$m) than clayey sediments, but more conductive than the sandy sediments of the substratum (above 100 $\Omega$m).

Variations of the exposure rate correlate with peat thicknesses found in boreholes, but thickness estimates could not be derived from the radiometric data without site-specific calibration. Therefore, the thickness estimates from the resistivity–depth models were used to scale the radiometric values. The deviation of resulting radiometric thickness estimates from the borehole results was also in the order of a few decimeters, but—compared with the electromagnetic results—the interface values undulated stronger, particularly close to the edges of the bog and where the exposure rate was extremely low.

In particular, the combination of both airborne methods yielded results that are (with a few exceptions with deviations of more than 2 m at about 5% of the boreholes) close to peat thicknesses found in boreholes (zero mean with about a meter standard deviation). A portion of these differences may result from the comparison of point values (borehole data) with grid values (airborne data), which are always smoother.

This case study focused on a single bog. Therefore, a more general investigation of numerous and diverse peatlands (both peat bogs and fens) will be necessary to prove and expand the approach presented here, which enables large-scale peat volume mapping without an imperative necessity of borehole data.

**Supplementary Materials:** The following are available online at http://www.mdpi.com/2072-4292/12/2/203/s1: Zip file Z1: Borehole databases including all data used in this paper at borehole locations and flight lines crossing the bog area.

**Author Contributions:** Conceptualization, B.S.; methodology, B.S. and M.I.-v.S.; software, B.S. and M.I.-v.S.; validation, B.S., M.I.-v.S. and S.F.; formal analysis, B.S. and M.I.-v.S.; investigation, B.S., M.I.-v.S. and S.F.; resources, B.S., M.I.-v.S. and S.F.; data curation, B.S. and M.I.-v.S.; writing—original draft preparation, B.S.; writing—review and editing, B.S., M.I.-v.S. and S.F.; visualization, B.S. and M.I.-v.S.; supervision, B.S. All authors have read and agreed to the published version of the manuscript.

**Funding:** This research received no external funding.

**Acknowledgments:** The authors would like to thank LBEG for providing the borehole data as well as the three anonymous reviewers for their comments, the Guest Editors and the Remote Sensing editorial team for helping to improve this manuscript and for enabling fast publication.

**Conflicts of Interest:** The authors declare no conflict of interest.

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
