# Peer review of "Airborne Electromagnetic and Radiometric Peat Thickness Mapping of a Bog in Northwest Germany (Ahlen-Falkenberger Moor)"

_remotesensing, doi:10.3390/rs12020203_

Round 1

Reviewer 1 Report

This is an interesting paper, which reports on two geophysical methods for measuring peat depth over a German bog.

The approach seems to be promising. However, I do have some major concerns with the organization and writing of the manuscript. The methodology is poorly described by the authors and very difficult to understand. As a result, my recommendation is major revision.

The manuscript can be much better organized. As presented now, the authors mix methods and results throughout  the entire paper. This is extremely confusing and limits the understanding of the message. The manuscript badly needs a road map showing how its various components are related to produce a robust proof of method.

I am sorry to say that the methodology is extremely poorly described and difficult to understand, partly because of the lack of organization of the paper, partly because the authors do no explain what they have done. This is a serious problem that prevents a comprehensive understanding of the paper and the findings and must be addressed very carefully in case the authors decide to resubmit the paper.

The Title should include the word “bog” so that the readers can understand what the paper is about. The Abstract lacks information on the specific findings of this paper and also needs to conclude with a statement on the significance of the study. The Methods section is badly organized and does not include enough information about the analyses performed by the authors. The Results section includes methods and information that are not described in the previous sections, which is unacceptable. The Discussion is short and does not get to the point. The Results section does not include any comment on the significance of this work.

Finally, in general, both terms "depth" and "thickness" are used. The authors should chose one of them and be consistent thorough the paper. The paper needs to be revised by a native English speaker because many sentences are poorly constructed and the sense of some sentences is difficult to interpret.

Line 18: I suggest to add a specific spatial scale

Line 69: Does the reference cited as (15) contain a different study from the one presented in this manuscript? In case it is the same study or the same study site/dataset, please rephrase this sentence explaining the difference between study (15) and this study. I suggest to remove the citation from this sentence and include it in the next sentence, describing what is the advancement that is shown in this new study.

Line 73: instead of “large-scale” please specify the scale.

Fig. 1: Please use a larger character for the scale bar. Also, include a legend for the airborne survey (polygon with the red parallel lines I guess) and the study site (black dotted rectangle), or include these specs in the figure caption. Finally, include an insertion with the location of the study site within Germany.

Fig. 4: substitute “Red lines outline the bog” with “The red line outlines the bog”

Fig. 4 line 115:  delete the parentheses for d. I do not see where are the “black circles”. Please specify and show in the figure.

Lines 120-126: Readers of Remote Sensing journal may not be familiar with these three different technologies, so a short description is needed for each one. Please also explain/describe figure 5. Moreover, I do not understand why you describe here the cesium magnetic method and then data provided by this sensor are not used in the analyses. Please clarify, and either delete the description of the HMG or include the analyses results.

Fig. 5: did you create this figure specifically for this paper? Otherwise, please cite the source.

Section 2.2.2: The structure of this section is very confusing. I suggest to separate the description of the three different sensors/data processing, splitting the Data processing section into three sub-sections, one for each sensor/technology.

Lines 136-137: delete "sophisticated" and describe in details the tools and the methodology. Please add a detailed description of the parameters used in the inversion. Did you use a constrained inversion? Specify.

Line 150: Please explain what Oasis montaj is. It is unacceptable that the readers should download another paper in order to understand the methodology that has been applied.

Fig. 6: This figure is incomplete and difficult to read. The y axis of all plots has too small characters and difficult to read. Please add intermediate numbers between the extreme values. It is not clear the position of the values along the y axis in all plots, markers are needed along the y axis. Also, in the text and in this figure you mention a “road”; what road? There is no road in the map shown in Fig. 1. I understand that these profiles correspond to the transect L94.1, however it is difficult to visualize the correspondence of the profiles to the features described in the map. I suggest to add one more transect that includes the profile of the soil surface, showing the soil elevation and other features (road, marsh, etc.). Moreover, what are the lines shown in the figure? There is no legend and in the figure caption there is no explanation of the different lines shown in the figure.

Lines 153-154: please explain what “half-space parameters”, “apparent resistivity” and “apparent depth” mean, so that the readers do not have to read the reference (40) in order to understand the methodology.

Line 163: what filtering procedure? Please explain.

Line 165: where do we quantitatively evaluate such correlation? Please modify this sentence in order to explain how the correlation has been observed in a quantitative way. Moreover, this sentence is poor, please rephrase.

Fig. 7: same comments as for Fig. 6. As in Fig. 6, also in this figure the y axis of all plots are not acceptable: the characters are too small, there are no clear markers corresponding to the numbers, intermediate values are needed (not just the extreme values). There is no legend that explains the different lines. I suggest to add the soil profile, as I suggested for Fig. 6.

Section 2.2.3: The peat thickness estimation (Lines 176-186) should be moved to the Results section, it should not be part of the Methods. I suggest to leave in the Methods section the modelling study - that I suggest to rename "sensitivity analysis". Why such analysis is only performed for the HEM data and the HRD data? Please add a similar sensitivity analysis also for the HMG data (in case you decide to show the analyses of these data too).

Line 181: substitute “will be” with “is”

Lines 195-198: This sentence is poorly formulated. Please check the English and the structure, using shorter sentences, one for each model case. Please specify the ranges of depth and resistivity for each case scenario.

Model 1 and Model 2: What is the resistivity of peat? Do you have field or lab resistivity measures? Or please refer to previous studies. This exercise is pointless if you do not specify what is the resistivity range that (likely) corresponds to peat. Model 2 for example shows that HEM detects well highly conductive materials, with resistivities between 1 and 5, but this range does not correspond to peat but clay. A discussion about the resistivities of different materials should be also added to the Discussion section.

Fig. 8: I and Q correspond to In-phase and Quadrature I guess. They should be defined in the text BEFORE you use them in a figure. The dotted line corresponding to the steepest-gradient analysis is difficult to distinguish. The misfit q should be defined in the text before than it is included in the figure caption.

Line 220: analysis instead of analyses

Lines 254-257: Have you tried different saturation values for example? You mentioned "realistic conditions for peatlands". Do you have specific measurements/data that show such realistic values for the study site? General comment on the HRD method: Given your results, I don't see how this methodology can be used to separate a peatland from a water body (i.e. nearby lake/pond for example). I think this should be discussed in the Discussion section.

Fig. 11 and section 3.1.1: I do not understand this figure. Is this referred to the top layer? How deep?

Line 285: A correlation between peat thickness and soil elevation has not been described in the Methods nor in the Introduction, so it cannot be included in the Results. In case you want to add such correlation, further analyses must be described in the Methods and results on the existence of such correlation must be included earlier on in the Results section.

Line 296: Vertically? Figure 11 does not include any info about the depth of the retrieval. Please see my comment above.

Lines 296-297: this part must be moved to the Methods section.

Line 301: I don’t recall you mentioned that the instrument has a laser altimeter on board. Please add this info when you describe the instrument.

Lines 303-306: This should not be explained here but in the Methods section.

Lines 323-325: please add a figure with measured vs simulated depths for all 97 boreholes. Include R2 and the root mean squared error.

Fig. 14: please explain all the terms included in the legend

Lines 340-348: separate the description of the applied method (that should be moved to the Methods section) from the results.

Lines 380-381: what does “combined” mean? First of all this part should be moved to the Methods. Did you average the values? Why you selected 2m as threshold? Did you weight the importance of one of the two components (HEM and HRD) somehow? What are your choices based on?

Line 405: sorry, I do not follow this comment. Random distribution is highly desirable for checking the error. I think the meaning of your sentence was intended to be different. Please rephrase.

Lines 407-410: what do you mean? boreholes are point values. The construction of this sentence is poor. Please clarify.

Line 442: the description of the instruments must be moved to the Methods.

Lines 444-450: I strongly suggest the authors to produce a comprehensive analysis of the deviation between the altimeter and the DEM data. Is it a systematic error? In case it is, altimetry could be easily corrected based on the DEM values. The use of corrected heights might reduce the error also of HEM data.

Line 458: how did you calculate this volume? Please specify the resolution.

Lines 463-464: the objective is more than estimate the lateral and vertical extent of the bog. I believe the authors should state in this first sentence that they are exploring to what extent the use of electromagnetic and radiometric data may be successful for peat detection.

Reviewer 2 Report

In this paper, airborne electromagnetic and radiometric measurements are used to estimate the lateral extent and the thickness of a bog in northwest Germany. The authors describe in detail the methods applied and all the steps used to process and refine the available data sets. They also point out the main limitations of such techniques and the potential sources of error that may have affected the obtained results. This is a very good paper that provides useful information to readers interested in quantifying peat volumes with airborne geophysical methods. It is well suited for the journal and deserves publication with no significant corrections.

I have only few minor specific comments:

Figure 1: It would be good to add an inset showing the location of the study area in Germany. In figure caption, explain the different types of colored lines.

Figure 2: this figure is quite confusing due to the small font size used. If possible, use different colors for each “unit” or increase font size. Anyway, this is a Quaternary geology map or a Superficial deposits map more than a “Geology” map.

L104: missing “)”.

L111: the sentence is missing a part (the ground-water level is at … ). Please amend.

Reviewer 3 Report

There is no comment and suggestion. The paper can be published as it is.

Author Response

No Response necessary.

Round 2

Reviewer 1 Report

Dear Dr Siemon,

it has been a pleasure to review your manuscript. The second version has greatly improved in my opinion, and the work is now ready for publication. I only have some minor suggestions. In the Introduction (Line 39) I suggest you to add at least one sentence about the conservation policies, I believe this might increase the description of the significance of your work.

Lines 70-71, I just want to mention that another paper on this topic has been recently published on JGR: "Quantification of peat thickness and stored carbon at the landscape scale in tropical peatlands: A comparison of airborne geophysics and an empirical topographic method".

Fig. 3: What is NN? Does it correspond to the mean sea level? Please specify in the figure caption. Moreover, the quality of this figure is very poor, it is hard to read the characters. I suggest to increase the resolution or over-wright the words somehow.

Line 131: In my opinion, there is no need to mention the magnetic data if then they are not analyzed. I believe that this is confusing for readers. However, as I said, this is just my opinion.

i think that the new structure of the paper is very clear, and I really appreciated the detailed explanations you provided about the methodology: this is important for the readers of RS journal.

I am pleased to suggest the publication of this work.
